# Facilitators and barriers to engaging in expressive writing among health and social care professionals

Michael A. Smith[1]*, Lauren M. Hoult[1], Daniel Rippon[1], Nicola O'Brien[1], Dawn Branley-Bell[1], Lucie Byrne-Davis[2], Caitriona Collins[3], Stephen Gallagher[4], Gail Kinman[5], Arron Mark[6], Daryl B. O'Connor[7], Kavita Vedhara[8,9], Glenn P. Williams[1], Mark A. Wetherell[1]

**1** Department of Psychology, Northumbria University, Newcastle upon Tyne, United Kingdom, **2** Division of Medical Education, School of Medical Sciences, University of Manchester, Manchester, United Kingdom, **3** Newcastle upon Tyne Hospitals NHS Foundation Trust, Royal Victoria Infirmary, Newcastle upon Tyne, United Kingdom, **4** Department of Psychology, University of Limerick, Limerick, Ireland, **5** Department of Organisational Psychology, Birkbeck University of London, London, United Kingdom, **6** West London NHS Trust, London, United Kingdom, **7** School of Psychology, University of Leeds, Leeds, United Kingdom, **8** School of Psychology, Cardiff University, Cardiff, United Kingdom, **9** School of Medicine, Keele University, Newcastle upon Tyne, United Kingdom

* michael4.smith@northumbria.ac.uk

## Abstract

Healthcare workers in the UK report high levels of burnout and poor wellbeing, and interventions are urgently needed to address this issue. Expressive writing, whereby individuals write about emotionally laden thoughts or experiences, is an effective intervention for reducing stress and enhancing wellbeing, and is a potential candidate for use with healthcare workers. However, it is crucial that the preferences of healthcare workers are taken into account in the design of future trials in this area, and that perceived barriers and facilitators to engaging with expressive writing activities are considered. Therefore, the present study aimed to seek healthcare workers' views on expressive writing activities for enhancing wellbeing. We informed 11 UK-based health and social care professionals about current research into the potential wellbeing benefits of expressive writing activities and conducted 1:1 semi-structured interviews to determine their preferences, perceived facilitators and barriers to engaging routinely with expressive writing. Data were analysed using deductive, reflective thematic analysis, with the coding framework informed by the Capability-Opportunity-Motivation-Behaviour (COM-B) Model. Key findings, from the perspective of future trial design, were that participants preferred the three good things (writing down three good things that have happened during the day), written benefit finding (writing about positive emotions, thoughts, feelings and life changes in relation to an unfavourable experience) and gratitude letter (writing a letter of appreciation to someone who has never been properly thanked) activities. Participants expressed preferences for simple, brief activities that could be easily embedded into

**Data availability statement:** All interview transcripts are available from the Open Science Framework https://osf.io/zfmtk/. (DOI: 10.17605/OSF.IO/ZFMTK).

**Funding:** The author(s) received no specific funding for this work.

**Competing interests:** I have read the journal's policy and the authors of this manuscript have the following competing interests: Michael A. Smith and Mark A.Wetherell have run expressing writing workshops which have generated income to Northumbria University. This does not alter our adherence to PLOS ONE policies on sharing data and materials.

a daily routine. However, it was clear that support from managers, researchers or friends and family would be needed to promote engagement with the activities. Most notably, participants expressed a preference for flexibility in terms of how, when and where they write.

## Introduction

There is a major workforce crisis in the UK health and social care sector [1]. This is driven in part by low morale and poor staff wellbeing, in turn leading to high staff turnover and chronic absenteeism [2]. Consequently, the quality of care and the experience of care recipients suffer [3,4]. A whole system approach is required to address this problem, with initiatives at sector, organisational, team and individual levels, given the substantial structural and environmental stressors that can adversely influence wellbeing in the health and social care sector [5]. Indeed, the National Health Service (NHS) in England has recognised the need to prioritise workplace wellbeing [6]. A rapid evidence review identified a variety of individual-level interventions that are being implemented to address the mental health of healthcare staff, including mindfulness-based stress reduction (MBSR) and resilience training [7]. However, it is acknowledged that the evidence base for the efficacy of existing workplace interventions in a healthcare context is limited [7]. Further, with respect to the implementation of workplace health and wellbeing interventions in the NHS, staff engagement is a notable barrier, with staff reporting that there is a lack of available time to engage with available interventions, and that they are difficult to access [8]. Thus, it is critically important to understand staff perceptions of barriers to engagement with any potential occupational wellbeing intervention, as this is likely to impact upon implementation success.

A potentially useful tool for enhancing psychological wellbeing among health and social care workers is expressive writing. Expressive writing activities involve writing about emotionally laden thoughts or experiences, and are known to convey a range of psychological and physical wellbeing benefits [9]. Further, expressive writing paradigms have an advantage over many other low-intensity psychological interventions because individuals can engage with them at no financial cost, with minimal training at a time and place convenient to them [10]. Therefore, they are potentially useful for reducing stress in individuals, such as frontline health and social care professionals, who typically provide care in dynamic settings with limited time and resources to engage in self-care during work hours.

Several different expressive writing paradigms have been proposed, including Written Emotional Disclosure (WED), whereby disclosing negative emotional experiences by writing about them has been associated with improved health and wellbeing [9,11]. However, other expressive writing techniques encourage individuals to focus on more positive aspects of their lives [12]. For example, in comparison to writing about non-emotive topics, writing about positive life experiences is associated with reductions in self-reported stress and anxiety in the general population [13]. Further,

it has been explored whether writing expressively about positive life experiences can convey psychological benefits in an online context (i.e., when participants 'write' by typing about their emotional experiences on a web-based portal). Previous work by Allen and colleagues [10] has found that in purposively recruited participants with relatively high levels of psychological distress at baseline, positive writing in this context reduces depression and perceived stress reactivity, relative to neutral writing. In a further online study, a beneficial effect of positive writing on anxiety and aspects of job satisfaction was observed in full-time workers [14]. A briefer and more acute positive expressive writing activity is three good things. This activity involves writing down three good things that have happened during the day, and has been associated with reductions in burnout and depression among healthcare workers [15].

Another positive expressive writing intervention is written benefit finding, which requires participants to focus on, by writing about, positive emotions, thoughts, feelings and life changes concerning an unfavourable experience. In a previous study, written benefit finding during the COVID-19 pandemic lockdown was found to reduce self-reported levels of state anxiety [16]. Further, there is evidence that this technique can attenuate anxiety in informal parental caregivers of children with autism [17]. Thus, it seems that providing carers with the opportunity to refocus on more positive aspects of their lives could promote wellbeing, and it is possible that such an intervention may be effective in individuals working in caregiving professions. However, it has been suggested that further feasibility work is needed to ascertain the specific conditions under which this intervention is likely to be most effective [18].

Further expressive writing activities which have shown promise for improving aspects of psychological wellbeing include gratitude writing, whereby people are instructed to write about people and relationships they are thankful for [19]. However, a recent review concluded that the literature on the efficacy of gratitude interventions for improving psychological wellbeing "remains scant, fragmented and inconclusive" [20; p. 758–9]. Another technique involves writing about one's best possible self as if everything has gone as well for them as it possibly could, and the goals that will help them to attain this best possible version of themselves. This activity has been associated with enhanced positive affect in students [21]. Taken together, there exists a range of expressive writing techniques focussed on more positive writing topics. A recent systematic review concluded that despite the relatively poor quality of studies in this area, reasonably consistent benefits of positive writing interventions have been reported [12].

While expressive writing techniques convey promise as a means of enhancing wellbeing, their potential role as an intervention for frontline health and social care staff is less certain. Anecdotally, some of these techniques may be particularly beneficial for these individuals, who are prone to high levels of burnout [22], and lack time to engage in self-care [23]. Indeed, expressive writing benefits have been observed following writing for as little as two minutes per day, suggesting that limited time need not be a barrier to engagement with these activities [24]. WED may offer healthcare workers the opportunity to process stressful or difficult events which they experience during their working lives, while positive expressive writing activities may provide an opportunity for reframing and focusing on positive aspects of their roles. However, only a small number of studies have investigated the efficacy of expressive writing techniques for enhancing healthcare staff wellbeing [e.g., 15,25]. Feasibility studies have shown that there are issues with written benefit finding for informal carers [18], and positive writing for professional carers [26] in terms of participants reporting having a lack of time to engage with expressive writing. However, qualitative comments reported by healthcare workers in Sexton and Adair's [15] study convey an overwhelming enthusiasm with the three good things format, suggesting that this briefer exercise may be preferred by participants as it is perceived as a more feasible technique to engage with. Additionally, further questions remain concerning the design of future feasibility trials of expressive writing for frontline health and social care staff. These questions relate to how often and for how long participants would want to write, whether participants would prefer to type or handwrite and whether there is a particular preference for any of the techniques described above. Furthermore, there may be other perceived facilitators and barriers to engaging with the techniques that could impact the efficacy of expressive writing for enhancing psychological and physical health among health and social care workers.

On this basis, we sought here to better understand how expressive writing practice can best be embedded in the daily lives of busy health and social care workers, using the Capability-Opportunity-Motivation-Behaviour (COM-B) model [27, see Fig 1]. The COM-B model comprises six sub-components positing that engagement with a behaviour is determined by an individual's capability (physical capability and psychological capability to engage in the behaviour, such as knowledge and skills), opportunity (social opportunity and physical opportunity, external to the individual) and motivation (automatic motivation and reflective motivation that make an individual want to engage with a behaviour, or not). Given the importance of these components in determining whether a behaviour will be sustained, it is critical to elicit beliefs of health and social care workers surrounding capability, opportunity and motivation to engage with expressive writing activities. Healthcare workers were informed about a range of expressive writing techniques and, in 1:1 semi-structured interviews, we sought their views on the different expressive writing approaches and their perceived facilitators and barriers to engagement. It has been established that staff wellbeing interventions in healthcare settings are most effective when staff are involved in their development and implementation [28], thus the present study represents important public engagement work to support this objective.

## Methods

### Design

A qualitative interview study design was employed to elicit barriers and facilitators to engaging with expressive writing, and to map these onto the COM-B model. This method was chosen to enable an in-depth exploration of participants' barriers and facilitators to engaging with expressive writing, and to gain an understanding of individuals' subjective experiences to ascertain how best to deliver an expressive writing intervention to healthcare workers. The study was pre-registered, prior to data collection, on the Open Science Framework (https://osf.io/zfmtk/).

### Participants

We aimed initially to recruit 10 individuals employed in the frontline health and social care workforce. This was aligned with the upper number of participants required for a small qualitative study, using 1:1 interviews and analysed using thematic analysis [29]. This number of participants is also adequate for deductive thematic analysis, when it is desired to observe at least one instance of each theme, where each theme is endorsed by 15% of the population [30]. In total, 11 participants (nine women, two men) were recruited to the study, ranging in age between 24 and 44 years. All participants worked in frontline health and social care within the NHS in England, although working for the NHS was not a prerequisite for taking part. Participants were recruited via the professional networks of the researchers and recruitment adverts on Twitter/X. Participants were paid £25 to reimburse them for their time. Although participants residing anywhere in the

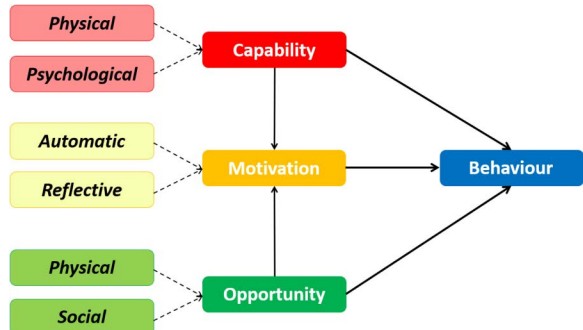

**Fig 1. The Capability-Opportunity-Motivation-Behaviour (COM-B) model [adapted from 27].**

UK were eligible to take part, all participants lived and worked either in London or North East England. Length of service ranged between two months and 8 years. Most participants worked in psychological or mental health – i.e., job roles included Psychological Wellbeing Practitioner (PWP), Cognitive Behavioural Therapy (CBT) Therapist and Clinical Psychologist. Other job roles included Midwife Sonographer, Physiotherapist and General Practitioner (GP). Some participants (n = 4) had experience in using expressive writing techniques, others (n = 3) were familiar with expressive writing techniques but never used them, while four participants had never heard of expressive writing. Participant details are summarised in Table 1.

## Procedure

All participants gave written informed consent prior to taking part in the study. The study was conducted in accordance with the Declaration of Helsinki, and the protocol was approved by the Ethics Approval Process of Northumbria University (reference 2839). The procedure was preregistered on the Open Science Framework prior to data collection (https://osf.io/zfmtk/).

The interviews were conducted via Microsoft Teams. Interviews were conducted between 15th March and 20th April 2023. All interviews were recorded, and transcripts were auto-generated within Microsoft Teams software. A semi-structured interview schedule guided the discussion in each interview. Participants were first asked about aspects of their job role that they find challenging or stressful, and were then asked to detail any activities or techniques they use to manage their work-related stress. A slide deck was subsequently shared with participants (see https://osf.io/82yj3) and delivered an approximately 10-minute presentation which provided a general definition of expressive writing, and then provided a specific description about different expressive writing techniques with examples of instructions for each activity. The slide deck summarises each of the expressive writing activities which were discussed during the interviews. In addition to the flagship WED activity [9], five positive expressive writing activities identified in a recent systematic review were described to participants [12]. A description of these techniques can be found in Table 2. Participants were told that they were free to interject at any point in the presentation to seek clarification, ask questions or provide comments, but no participants did this. Following the presentation, participants were asked about their perceptions of the acceptability and efficacy of the expressive writing techniques (e.g., potential benefits, burdens, preferences for any specific techniques, any difficulties that might be encountered and whether colleagues would be likely to hold similar perceptions). They were also asked about barriers and facilitators to implementation of the activities in day-to day life, including i) beliefs about

**Table 1. Details of all participants (age, gender, location, job role and time in job role) and familiarity with expressive writing techniques (unfamiliar = never heard of expressive writing techniques before, familiar = have heard of some techniques before but never used them, experience = experience using at least one of the techniques) of all participants who took part in the study.**

| Participant | Age | Gender | Location | Current job role | Length of service | Expressive writing familiarity |
|---|---|---|---|---|---|---|
| 1 | 32 | Male | London | PWP | 17 months | Experience |
| 2 | 26 | Female | London | PWP | 2.5 years | Familiar |
| 3 | 27 | Female | London | CBT Therapist | 3 years | Familiar |
| 4 | 37 | Female | North East England | Midwife Sonographer | 8 years | Familiar |
| 5 | 31 | Female | London | PWP | 3 years | Unfamiliar |
| 6 | 24 | Female | North East England | Physiotherapist | 2 months | Unfamiliar |
| 7 | 44 | Female | North East England | Physiotherapist | 5.5 years | Experience |
| 8 | 30 | Male | London | PWP | 2.5 years | Experience |
| 9 | 24 | Female | London | PWP | 2 years | Unfamiliar |
| 10 | 36 | Female | North East England | GP | 5 years | Unfamiliar |
| 11 | 36 | Female | North East England | Clinical Psychologist | 4 years | Experience |

**Table 2. The expressive writing activities discussed during the interviews and the descriptions provided to participants.**

| Activity | Description |
|---|---|
| Best Possible Self | Write about your life in the future as if everything has gone as well as it possibly could. |
| Gratitude | Write a letter of gratitude to someone who changed your life for the better or who you never properly thanked or write about an aspect of life for which you're grateful. |
| Positive experiences | Write about a positive experience or issue that has affected you and your life. |
| Three Good Things | Recall and imagine three good things that went well today; think of an explanation for each good thing; after creating a mental image, write down each experience. |
| Benefit finding | Write about the benefits of a stressful or upsetting experience. |
| Written Emotional Disclosure | Write about an extremely upsetting, stressful or traumatic experience. |

training or education that would be needed, ii) the influence of other people (e.g., friends and family) on engagement, iii) home and work environment influences, iv) incentives and v) ease of engaging. Participants were then specifically asked how often they would engage, how long they would want to write for, any preferences around the mode of writing (e.g., writing by hand or typing) and whether there were any preferences around location (e.g., at home or at work). Finally, they were asked about their perceptions on the extent to which the techniques could address the wellbeing needs of health and social care workers. Following each interview, upon review of the recordings, transcripts were amended for accuracy. Full transcripts (with identifying details removed) are available at https://osf.io/zfmtk/.

## Analysis

Interview transcripts were imported into QSR International NVivo Pro 12 software. Each time a participant endorsed a specific technique by expressing their interest in using it, discussed their preferred writing duration or preferred mode of writing (i.e., writing by hand or typing), this was recorded, and for each domain, a total number of mentions was computed. It was possible for participants to endorse multiple options within each domain (e.g., some participants expressed a preference for multiple writing techniques, and some endorsed both writing by hand and typing as preferred modes of writing).

Data were analysed using reflexive thematic analysis, taking a predominantly deductive approach [29]. Firstly, two researchers (MS and LH) independently coded the data from one participant. At the first level, data were coded with respect to whether a barrier or facilitator to engagement was being discussed. At the next level, data were coded according to alignment with each of the six COM-B constructs, which served as the deductive themes. Subsequently, coding labels were generated within each COM-B construct as the coding process evolved to identify clusters of similar data. Once this process was completed for a single participant's data, the researchers met to discuss the codes and to consider any similarities and differences in the coding. A consensus was reached, which then informed the coding of the remaining data. Subsequently, these two researchers (MS and LH) independently coded the data for the remaining 10 participants. These two researchers then shared coding schemes for review and subsequently met to refine the codes, establish a consensus, and discuss the best way to present the themes. MS, LH, DR, DBB, CC, SG, AM, NOB, DO, KV, GW and MW then met to discuss each theme and related codes. MS discussed the themes separately with LBD and GK.

These meetings allowed all authors to discuss their interpretation of the data, to further refine the codes and consider the best way to present the data. Following these contributions from all authors, the analysis was finalised by MS.

### Reflexive statements

We acknowledge that the reflexive thematic analysis approach taken here is underpinned by the subjectivity of the researchers. It is acknowledged that, as active agents in the research, our experience is important in shaping the work and making meaning of the data. The outcomes generated from the analysis are the result of collaboration between the participants and the authors who undertook the analysis. Herein we provide reflexive statements for the two authors who were primarily responsible for undertaking the analyses.

MS is a white, middle-class, cisgender, straight, non-disabled male, resident in the UK, and who grew up in a working-class household and neighbourhood in Australia. MS is an NHS service user, but also has private health insurance. MS is an academic psychologist, conducting teaching and research primarily in health psychology and psychobiology. MS teaches undergraduate students about the COM-B model and uses this model in his research. MS has conducted several studies on the psychological and physical health benefits of expressive writing and has delivered workshops on this topic in a variety of contexts, including to staff working in healthcare settings. MS has never worked in the healthcare sector but has several family members and close friends who work in this sector, including for the NHS.

LH is a white, middle-class, cisgender, straight, non-disabled female, UK resident, NHS service user, and who grew up in a working-class household in the North-East of England. At the time of data collection and analysis LH was a PhD researcher studying positive expressive writing interventions for improving wellbeing. Her research implements various methodological approaches to investigate how positive writing interventions should be delivered, the populations they work for, and the health outcomes they most reliably benefit. LH completed an MSc in Health Psychology and acquired knowledge of the COM-B model theory and its practical application within research contexts. LH has experience volunteering at an NHS rehabilitation and recovery unit service for individuals with complex mental health needs and has several close friends who work for the NHS.

## Results

### Reported stress and coping mechanisms

Participants were asked whether they were currently experiencing any work or non-work stressors. One participant was on leave from work due to stress. Additionally, seven participants reported current stress due to overwork (admin, high turnover of patients, lack of time for breaks), seven participants reported stress due to role responsibilities (identifying and managing risk, managing expectations, decision making, dealing with patient complexities), one participant reported stress due to low pay and one participant reported stress due to there being no mechanisms for debrief or reflection following incidents at work. Participants were additionally asked whether there are any activities they engage in to cope when feeling stressed. Activities reported were exercising (n = 7), socialising (n = 6), setting boundaries to maintain work-life balance (n = 5), formal or informal reflection/debrief following work stress (n = 3), engaging with nature (n = 3), practicing mindfulness or meditation (n = 3), recreational activities (e.g., cinema, reading; n = 3).

### Preferred activity

Most participants stated that their preferred activity would be three good things (mentioned by nine participants = 82%), followed by gratitude letter and written benefit finding (both mentioned by eight participants = 73%). The activity with the lowest number of endorsements was best possible self (mentioned by two participants = 18%; see Fig 2).

## Preferred mode of writing

Most participants stated that their preferred mode of writing would be writing by hand (mentioned by ten participants = 91%). Only three participants (27%) endorsed typing as a mode of writing (see Fig 3).

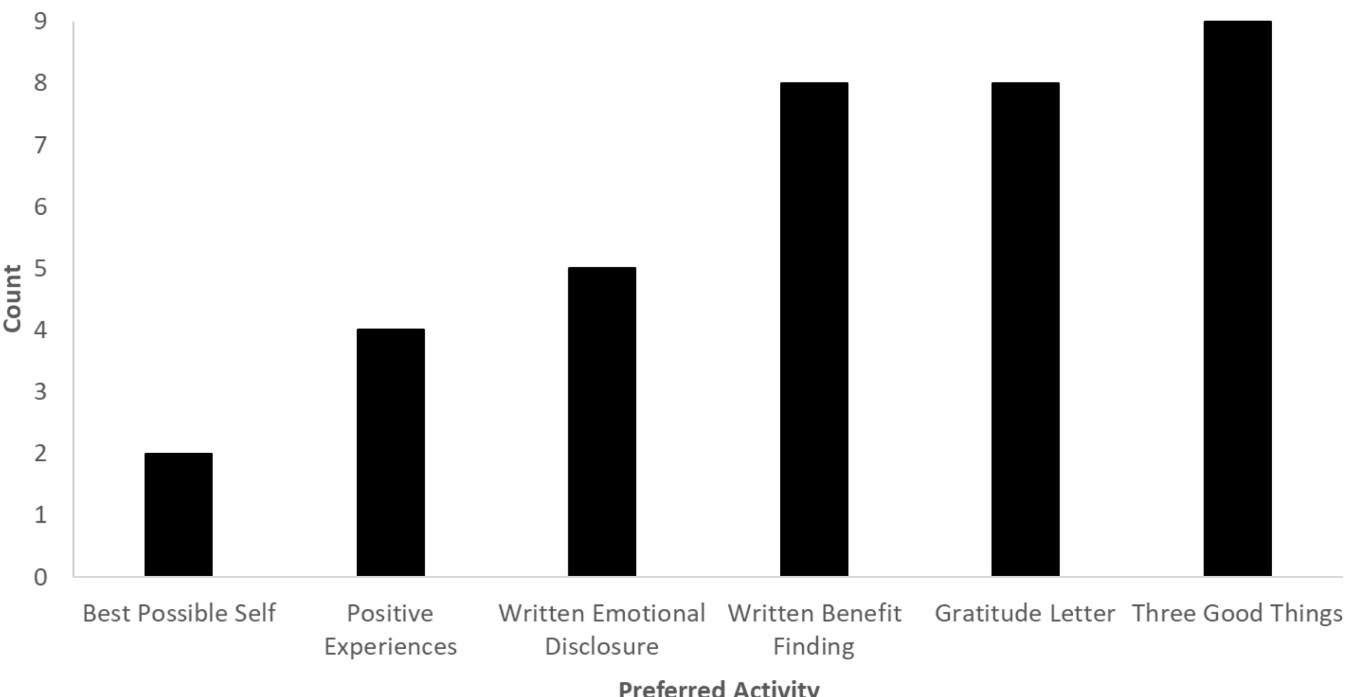

**Fig 2. Participants' preferred expressive writing activity.** Three good things was endorsed by the majority of participants, followed by gratitude letter and written benefit finding. Best possible self was the least endorsed activity.

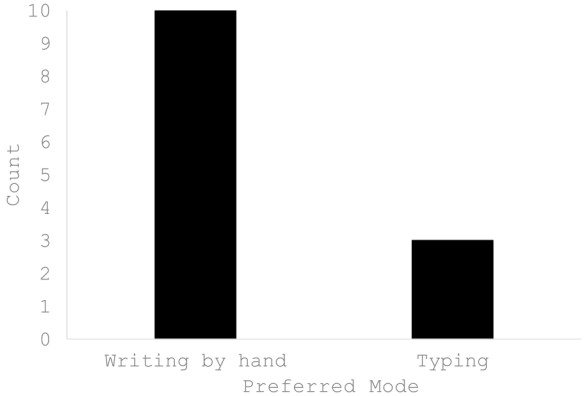

**Fig 3. Participants' preferred mode of writing.** Participants endorsed writing by hand as their preferred mode of writing over typing.

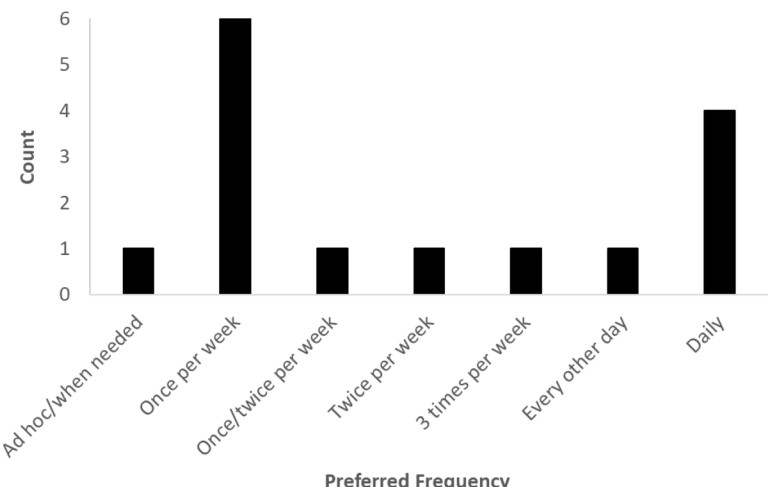

**Fig 4. Participants' preferred frequency of writing.** Writing on a weekly basis was the most strongly endorsed preference, followed by daily writing.

### Preferred frequency

There was no clear consensus with respect to how often participants would prefer to write. Six participants (55%) stated that their preference would be to write once per week, while four participants (36%) expressed a preference for daily writing (see Fig 4).

### Preferred duration

There was no clear consensus in terms of the amount of time that participants would prefer to write in a single session. Their preferred duration was largely dependent on the writing activity they would be engaging with and the frequency with which they would write. Some participants preferred the approach of writing little and often, whereas others stated that they would prefer to write for longer but less frequently.

### Summary of COM-B subthemes

On each occasion that participants mentioned a facilitator or barrier to engaging with expressive writing, this was coded according to the COM-B construct with which it was aligned, and subthemes were created. A total of 21 subthemes were identified, encompassing all six sub-components of the COM-B. Nine subthemes were identified as both barriers and facilitators depending on context: familiarity, perceived simplicity or difficulty, beliefs about impacts, mode of disclosure and debrief, minimising distractions, social support, conducive environment, dedicating time for writing, and optimal instructions for self-reflection. Eight subthemes were identified as facilitators: structure and prompts, intentional engagement, feeling the benefits, knowledge of evidence supporting benefits, monitoring benefits, rewards, flexibility and variety, and mode of choice. Finally, four subthemes were identified as potential barriers: learning difficulties, remembering to engage, not one-size-fits-all, and fails to mitigate structural workplace stressors (see Fig 2). Participants most strongly endorsed facilitators and barriers aligned with the reflective motivation component (199 mentions), followed by the physical opportunity component (151 mentions). No participants endorsed any facilitators aligned with the physical capability component of the COM-B model (0 mentions, see Table 3). A thematic map is presented in Fig 5. The subthemes aligned with each COM-B construct are described in further detail herein.

**Table 3. Number of mentions of a facilitator or barrier to engagement with expressive writing, coded against each COM-B sub-component. Facilitators (F) and barriers (B) were aligned mostly with reflective motivation and physical opportunity.**

| COM-B component | Sub-component | Subthemes | Total codes |
|---|---|---|---|
| Capability | Physical | Learning difficulties (B) | 1 |
| | Psychological | Structure and prompts (F) | 36 |
| | | Perceived simplicity or difficulty (F, B) | 20 |
| | | Familiarity (F, B) | 10 |
| | | Remembering to engage (B) | 3 |
| Motivation | Automatic | Not one-size-fits-all (B) | 9 |
| | | Feeling the benefits (F) | 8 |
| | | Intentional engagement (F) | 6 |
| | Reflective | Beliefs about impacts (F, B) | 119 |
| | | Mode of disclosure and debrief (F, B) | 44 |
| | | Fails to mitigate structural workplace stressors (B) | 24 |
| | | Knowledge of evidence supporting benefits (F) | 8 |
| | | Monitoring benefits (F) | 2 |
| | | Rewards (F) | 2 |
| Opportunity | Physical | Dedicating time for writing (F, B) | 81 |
| | | Conducive environment (F, B) | 46 |
| | | Flexibility and variety (F) | 12 |
| | | Mode of choice (F) | 9 |
| | | Optimal instructions for self-reflection (F, B) | 3 |
| | Social | Minimising distractions (F, B) | 27 |
| | | Social support (F, B) | 16 |

## Physical capability

**Learning difficulties.** Only one participant highlighted a potential barrier for physical capability in that individuals with dyslexia might find expressive writing to be difficult and unenjoyable, and therefore ineffective. Therefore, considerations should be made regarding the most suitable mode of delivery for such individuals.

*"I see a lot of people with dyslexia, I have a lot of friends with dyslexia. So is this something that they're able to do? Do they enjoy doing? How would they implement that?... When it comes to writing, are they physically able to do that? Do they get tired doing that? How do they find that (compared) to writing on the computer?" P1*

## Psychological capability

**Familiarity.** Participants discussed familiarity as an important factor to facilitate engagement. Some PWPs possessed a greater understanding of expressive writing techniques due to implementing them within clinical practice. Therefore, such participants were familiar with the techniques and perceived themselves as capable of engaging in expressive writing. For other participants, it was viewed that as they become more familiar with expressive writing techniques, they would begin to learn what works for them, which would make the activities both easier and more beneficial.

*"For me this feels like something you have to kind of learn by doing. I suppose you can improve it by your own way of doing it, and I suppose the way I've explained it is I've used it when it felt necessary. I write it in a handwritten way. I don't put any timers to it. I don't need to listen to music. I do it in my own home. I suppose, like these are all things that*

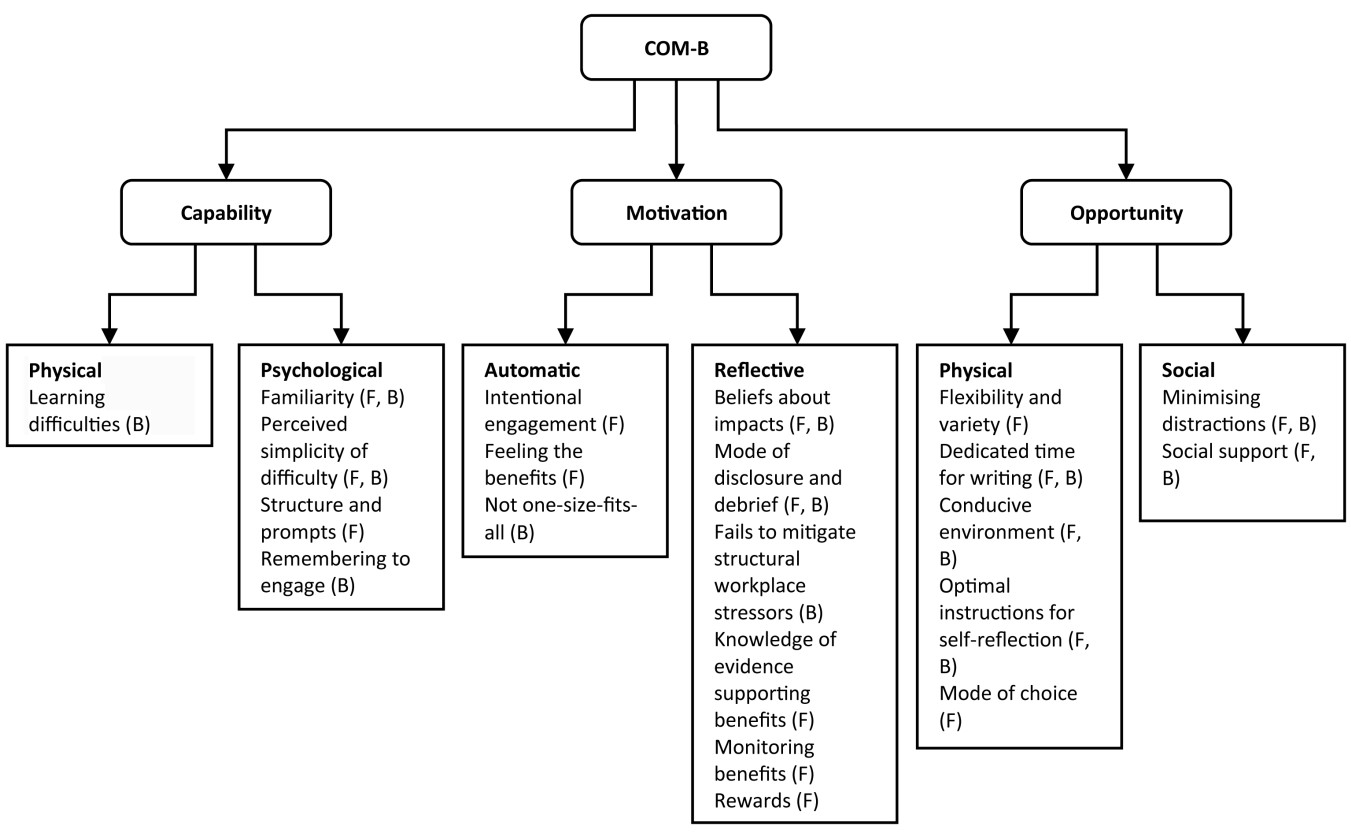

**Fig 5. Barriers (B) and facilitators (F) aligning to the subcomponents of the COM-B.**

*I've learned by doing it myself. And I suppose I think, only by doing it individual can work that those kind of preferences or way to improve it themselves." P8*

However, it was also suggested that unfamiliarity with expressive writing tasks could be a barrier in that they can be difficult to complete, particularly when just starting out.

*"I think once you start it and you find it helpful, that's easy then. But it's the initial starting point… That's a difficulty because it's preaching to the choir with me. I kind of like it. I guess it's really helpful, but someone who's never done it before… I could imagine that could be trickier. I suppose that's just an interesting thing to think about." P8*

**Perceived simplicity or difficulty.** Participants highlighted the need for any expressive writing techniques to be simple and uncomplicated.

*"It doesn't want to be complicated. Because, you know, we don't want to make it into a novel and overcomplicate things. And I think the simpler, the easier." P1*

However, when referring to specific writing techniques, some potential barriers were raised. Participants stated that some individuals might struggle to identify the positives when writing about three good things or benefit finding, despite knowing

the potential benefits, which could elicit feelings of guilt. Additionally, the ambiguity surrounding best possible self was proposed as a barrier, as it requires envisioning and describing an event that is yet to occur.

*"I think if you're not in a good place, sometimes I would be like well I can't even think of three good things that have happened today. What's wrong with me? You know I've got such a nice life and I'm so lucky and I'm so fortunate and I can't even find three good things I literally can't think of them." P4*

*"I'm not saying best possible self's not useful. But it's harder to come up with something, isn't it? 'Cause you're thinking about a situation that's not happened yet and I think that's probably a little bit harder." P7*

**Structure and prompts.** To improve simplicity and ease of use, participants recommended providing templates, prompts or examples to help them initiate and structure their writing, which would make it easier to engage with. For example, many participants endorsed the instructions and examples that were provided for each expressive writing technique during the 10-minute presentation. Additionally, some participants suggested that questions (e.g., *"what were you feeling at the time?"*, P2) could be a good way of structuring writing by splitting the writing into manageable chunks. These were highlighted as important to stimulate ideas and reduce cognitive burden, particularly following a busy working day.

*"I guess maybe just prompts for what to think about as well? Like sometimes when I get a bit brain dead (from) thinking all day so intensely that something that's gonna make it easier for me to think about it, to have prompts would make it easier." P2*

**Remembering to engage.** Some participants mentioned that it can be difficult to remember to engage and to establish a writing routine due to losing track of time during the day. Therefore, committing to engagement requires discipline and planning.

*"It's just that discipline, because I think I can see it from the view that this would be helpful. I do try and implement it where I can, it's just that kind of reminder, that discipline. So the barriers kind of, me and my time and do I remind myself to do this type of thing." P8*

### Automatic motivation

**Intentional engagement.** There was a belief among some participants that the frequency with which these activities are used shouldn't be too prescriptive, and that there would be an automatic motivation to engage with the techniques as and when necessary for the individual.

*"I suppose, maybe it's better to just pick it up and use it when you need it and not put the extra layer of a frequency that you have to dip into these techniques for them to be useful." P4*

**Feeling the benefits.** The majority of participants suggested that feeling benefits from the activities would be key to facilitating continued engagement with the activities. However, participants acknowledged that this would only incentivise continued use once they had begun engaging in expressive writing.

*"I suppose if I did it and actually I started to feel benefits or felt better by doing it then that would like encourage me to do it more." P6*

**Not one-size-fits-all.** It was discussed that expressive writing may not be a 'one size fits all' intervention. Participants stated that there are likely individual differences, which may mean that some activities are better suited to some individuals than others, or that some people might not benefit from any of the activities.

*"I think different ones would appeal to different people based on their personality, if that makes sense. I don't know if I would think that everyone would help me or what, or whether I would like actively do one." P6*

### Reflective motivation

**Beliefs about impacts.** Participants' beliefs about the potential positive or negative impacts were discussed by all participants as key facilitators and barriers to writing engagement. In terms of benefits, most participants were of the view that these techniques would be beneficial for reducing work-related stress and promoting more positive emotions in relation to work. Positive writing techniques were perceived as useful for reducing rumination in relation to work stress by shifting perspective to more positive and meaningful thoughts and increasing work satisfaction by recognising personal-strength and growth.

*"I think on days where you just feel a bit negative with work and you don't really feel like there's too many positive things in your day. It will help me shift my attention to the positive things." P2*

*"I definitely think at the times of my highest levels of stress and anxiety through work, this isn't something I've ever done in a in a work capacity. And I do think it would have been beneficial." P4*

Participants believed that WED would be beneficial for processing and rationalising stressful events at work, and that writing specifically would serve as a cathartic release.

*"I feel like if I look back to that event and maybe used a technique like that, where it didn't matter if it made sense, you know it didn't matter if what came out of the pen on the paper, it wouldn't matter if that made sense to anyone else or not. It would have been like a release for me. And I think that might have been helpful then." P4*

*"I wonder whether it would be quite useful if you do have an odd, now and then, negative experience or stressful time at home and you actually write your thoughts down about how you feel. And it's probably quite cathartic, isn't it, to do that." P7*

More generally, participants expressed a view that there would be mental health and wellbeing benefits which would incentivise continued engagement with the activities. For example, participants proposed that writing could be calming and enjoyable, could motivate and encourage them to achieve their goals, and potentially help with sleep if completed before bedtime.

*"It just helps you be a bit more positive on those down days and the best possible self, that sounds like a really, really nice one to help you feel motivated to achieve your goals. Like just thinking about where you could be and all these exciting things that you could achieve. And I think that would help me feel more motivated." P2*

On the other hand, there were some perceived negative impacts of certain expressive writing techniques, specifically best possible self and WED. Despite perceiving WED as potentially beneficial, some participants felt the technique could be daunting due to concerns that reflecting upon and writing about negative emotions could elicit distress. Participants also

stated that best possible self could highlight the gulf between their actual and ideal future selves, which could elicit negative emotions.

*"Writing down distressful situations can almost bring it to the forefront"* P1

*"I think again there's a flip side to that that (BPS) actually, in the here and now that might just make you feel further away from where you want to be"* P4

**Mode of disclosure and debrief.** Expressive writing was considered by participants to be an alternative, or an adjunct, to talking about emotions, as a means of disclosure.

*"Obviously talking about your experience, talking about your stressful situation helps, doesn't it? So you're kind of doing that, but you're writing it down rather than talking to somebody else. I think that would be helpful."* P7

This was highlighted as particularly important to promote self-reflection about stressful or traumatic experiences at work. Participants mentioned that in healthcare, there should be the opportunity to have a debrief following a traumatic experience at work (e.g., the death of a patient), but workplace demands often mean that this isn't possible. Therefore, expressive writing is a potential vehicle to enable this, where a formal debrief doesn't occur.

*"I'm thinking about the nursing staff that I work with on the unit at the hospital… there are a lot of staff that I work with that see a huge amount of traumatic experiences. And there's not always the space and the time to have a proper debrief. And this kind of seems like there would be, maybe, it ticks that box a little bit. It gives them a skill that they can use to help make sense of a really difficult traumatic experience in the hospital. And so, that's a really good reason for trying some of these things."* P11

However, participants described that it is potentially discomforting to commit thoughts and emotions to paper. Participants discussed how not liking or enjoying the process of writing would be a likely barrier to engagement.

*"It's quite like scary to have your thoughts out on paper and in front of you"* P5

*"I guess I've never been a big writer. I've never liked sitting and writing, never liked writing English essays, whatever. And so it's not something I do naturally anyway."* P10

**Fails to mitigate structural workplace stressors.** Participants raised the issue that there are several structural and organisational stressors within healthcare systems, which were perceived barriers to positive impacts from engagement. It was discussed that although expressive writing activities might be beneficial for helping individuals to cope with these structural workplace stressors, it wouldn't fully alleviate feelings of stress that are triggered by these organisational factors.

*"I know this is gonna sound weird, but let's say someone stabs you with a knife, right? This kind of technique, and this is what we do in therapy, what we're doing is we're stopping the blood flow coming out from that knife wound. But until that stressor, until that trigger of the stress is removed, the knife, i.e., better pay, better conditions, more staffing, you're always gonna be stressed because that knife you're always gonna be in pain because that knife is still in there. So this won't pull the knife out but what it will do is stop the blood coming out. It's helping you cope with things maybe a bit better, but your stress levels are still gonna be the same. And I think as human beings we sometimes get mixed up that coping with something makes you less stressed, but it doesn't. Coping with something helps you deal with high levels of stress, but it still means you're still quite stressed."* P1

As a result of workplace demands, participants highlighted the importance of rest time outside of work hours. Therefore, it was discussed that expressive writing could potentially feel like a chore that would interfere with leisure or relaxation time, and this could have a negative impact on the individual.

*"So, let's say if you've just finished work and it's been a hectic day, the last thing I would want to do is then sit there and write for instance, whether that be via phone or via laptop or by notepad. You kind of just wanna zone out and just do whatever helps you to relax or attend to your personal needs, etcetera." P5*

**Knowledge of evidence supporting benefits.** To facilitate engagement, one reflective motivator mentioned by participants was knowledge about the benefits of expressive writing. Specifically, participants mentioned that they would feel motivated to engage with expressive writing if they were informed about the research underpinning the techniques, which provides evidence for the potential benefits.

*"I think if there was more research to show how much it does help with wellbeing and with stress, I think that would push me further. And for instance, let's say if I'm doing the written emotional disclosure, maybe I'm thinking like, 'why am I doing this?' like 'I can't believe I'm doing this. I feel very exposed. I feel [a] bit raw, just putting it all out there'. But if I had previous or existing research to go by which said 'when you do engage with said technique, you know in the long run it does help with whatever.' I think that would be a good incentive for me to continue engaging with the technique." P5*

**Monitoring benefits.** To facilitate continued engagement, participants discussed the importance of helping people to see that the techniques are working over time as another facilitator for engagement. It was suggested that people could be encouraged to complete a mood scale periodically, to help them notice positive changes in their emotions, which could be a way of incentivising people to keep engaging with the techniques.

*"I'm not sure how you'd capture that. Maybe there'd be a measure you could use like an outcome measure or a scale or something to rate your mood or your happiness, and if you can see it's changing, maybe you'd be more incentivised to carry on." P7*

**Rewards.** Participants suggested that intangible rewards (e.g., watching television, seeing how engagement would benefit others) and tangible rewards (e.g., vouchers, certificates) would incentivise people to persist with using the techniques and may motivate engagement.

*"Maybe giving them a voucher at the end or something like a self-care voucher. One of the things I always say to my clients is you know, if you make it to the sixth week of doing this, I'm gonna give you a little certificate or something to say 'well done, you've done self-care' or something like that because you're not allowed to give the money on the NHS. But something like that. So it's like an incentive for them. So that's probably a good way to do it." P1*

## Physical opportunity

**Flexibility and variety.** When considering which of the techniques people would prefer to use, participants discussed that the different techniques would be beneficial in different contexts. For example, on particularly stressful days, a technique like WED or written benefit finding might be useful to manage stress, whereas on a more positive day, expressing gratitude might be a way of harnessing and further boosting positive emotions. Therefore, it would seem appropriate to offer people a suite of different techniques that they could use in different contexts.

*"I think it's down to personal preferences and I think also it's down to the kind of day you're having. Maybe I've had a great day at work, which I've had many of those and therefore maybe I don't really need to do benefit finding. Maybe I can just solely focus on three good things, for instance, or, maybe I can focus on gratitude because that day perhaps I found myself just thinking there's a lot of things that are not going well for me."* P5

**Dedicating time for writing.** There was a strong consensus among participants that having the time to write is a critically important facilitator. The importance of making time to engage with expressive writing was highlighted. Participants mentioned that making writing habitual and building it into their routine would support continuous engagement with expressive writing.

*"If there's a set time, then for me it's already planned and I know I've got that like, locked in to do then, and so then I won't book anything on top of that, for example. Or if things wouldn't work out on the set time that I would have it, I would just move things around and then make sure I've done it when I have my free time."* P3

*"If you want to build it into your routine like I say it needs to be a habit that you do religiously."* P4

Participants discussed the importance that some of the techniques are relatively brief (such as three good things) so that they could engage easily during a busy daily routine.

*"I think I would struggle more with the writing long-winded, I think writing's difficult actually. I think we don't write now do we, just we type, but I think I have always struggled with doing quite reflective pieces of writing where you have to write about your thoughts and feelings, but I could easily come up with three good things that have happened today or three things I'm grateful for that kind of thing… So, to me, the way you just think of three things, it's quite quick you just, you can fit that into your day."* P7

Participants stated that having reminders to complete expressive writing activities would promote regular engagement. These could be routine-based, where an individual chooses a particular time of the day or week where they write, by leaving writing materials or a journal in a prominent place such as the bed side table, or by setting an alarm.

*"I guess, how you could address time could be setting reminders or alarms and things to remind you to do it."* P9

However, participants discussed the difficulty of forming a routine where expressive writing becomes habitual, in part because of other demands on individuals' time.

*"By the time you get home at the end of the day and look after the kids and get them to bed and things like that, there's not so much time left in the day to sit down and do something like this."* P11

**Conducive environments.** Participants stated that to maximally benefit from the activities, they would need a calm and private location to write.

*"I think what would be most beneficial is a calm environment"* P4

*"I think making sure that you're in a confidential space to do it"* P3

With respect to both the availability of time and the timing of when expressive writing might be most beneficial, participants reflected on the possibility of engaging with expressive writing during the working day. However, participants

suggested that there would be several barriers to engaging with the activities at work, including a lack of time during the day, 'buy in' from management, and the physical work environment not being conducive due to a lack of private, quiet physical spaces.

*"There's just not huge amount of time allotted out for personal things. It is very much like back-to-back patient. So unless admin specifically allotted out time… There wouldn't be obvious periods of time that I could think of through the day where I would think, 'oh, I could just fit it in there'." P6*

*"I'd have to block time in the diary, which would then have to be justified and explained to management, because everything's checked in terms of each block of time that you're using needs to be filled with a patient. So that would be a problem at work." P7*

*"It's quite busy when I'm in the office at work. I think having a natural space to do it properly. And also when you're not in kind of a work mode either, I think it's good to be in a space where you do feel a bit more relaxed maybe? And you're not already thinking about work or what you've got to do next." P9*

Although a number of barriers were discussed with respect to engaging with writing activities during the working day, it was mentioned that the home environment can also be non-conducive.

*"At home there's no set desk or anywhere, it would be downstairs and kind of an open plan area where either there [would] be someone else there to distract me or I'll be conscious that there's a pile of washing or the kids toys need tidying or something else that would distract me and I'd put it off." P10*

**Optimal instructions for self-reflection.** Some participants indicated that the instructions for writing should allow literary freedom and self-reflection. While many participants were enthusiastic about the idea of prompts or examples to guide individuals in terms of what they could write about, it was highlighted that such prompts could interfere with spontaneous self-reflection.

*"I feel like it's good not to give too much of a prompt because it's good not to give examples because it's meant to be your own reflection. So I think if you see an example there or if you see too many prompts, you might be drawn down a different way rather than your own personal reflections." P9*

**Mode of choice.** Participants tended to have robust views about the mode of writing they would prefer to engage with, and it was clear that most participants would want the option of being able to write in their preferred mode, rather than being forced to write by hand or type. Although most participants endorsed writing by hand as their preferred mode, it was clear that if a participant strongly preferred one mode of writing over the other, then they likely wouldn't engage if forced to use a different mode.

*"I get the that if you were more that mindset, these type of techniques could easily be translated into the electronic world as well couldn't they. But I'm just a pen and paper type person and I think I've found whenever I've done it before I think having something in black and white in your own handwriting it just feels more personal." P4*

*"For me it would be on my phone because I always have my phone on me. Um, as much as I like the idea of, you know, having a dedicated journal, I just know I'm not really going to do it. Like, even if I get the intention to do it because I don't always carry my journal with me, not that I won't do it, but I will always have my phone on me so I can always just, you know, just whip out my notes and just log it there and reflect on it afterwards." P5*

### Social opportunity

**Minimising distractions.** Most participants highlighted the importance of having a quiet space and minimal distractions to engage in writing. Participants who live in a household with others, and particularly those participants with young children, highlighted the importance of having some time without distractions from others to enable them to immerse themselves in the writing activities.

*"I just have to find somewhere quiet and explain to the kids and stuff and maybe do it when they've gone to bed. Or on my day off which is today when they're at school." P7*

Distractions when living in a household with other people, such as living in shared accommodation or with young children, were strongly endorsed as barriers to engaging with expressive writing.

*"At home if it was noisy and I couldn't concentrate, that would stop me from doing it there and then." P3*

*"With three young kids, they're not gonna let me sit down and process it even for two minutes. A three-year old would be there going 'what you doing? Can I do it? Can I draw there?' or whatever, so as much as they would probably encourage it, I think they'd probably be a hindrance, unless I'd be in the house on my own, it would be a distraction or a hindrance from them." P10*

**Social support.** Participants highlighted support from others primarily as a facilitator to engagement in expressive writing. Many participants discussed the importance of family and friends, both for keeping them accountable to engage with the activities and to allow them the time and space to write without distraction. Additionally, participants stated that encouraging friends or family to engage with the activities as well might promote engagement.

*"Friends or family could help me keep accountable or if you were doing it with someone, that would help you. Or even if you just told them you were doing it would probably make me more likely to do it." P2*

*"It would be helpful for them to see, to understand why I'm doing it and what benefit there might be that I might gain from it. And, I guess if they're sort of on board with the idea behind it, then they may be more willing to facilitate me taking the time to do it or, reminding me to do it or encouraging me to do it." P11*

While participants discussed that support from friends and family would facilitate engagement with expressive writing activities, conversely, it was mentioned that unsupportive friends and family would be a barrier.

*"I'd imagine if they [friends and family] don't really see the benefit in it or if they speak negatively about what the benefits would be. If you just don't have time through the day, if they're asking you to do stuff in an evening…" P6*

Finally, participants discussed how support from their line manager, either by encouraging expressive writing practice or by freeing up time at work to complete expressive writing activities, would promote engagement.

*"I think it's something that I feel my supervisor, my line manager, the senior people within my department, I think they would be like, 'oh yeah this is it, this is a great well-being activity, you should, you should do it' and I think I could see it being promoted as a thing that as a department, we could all potentially be engaging in." P11*

## Discussion

Here, we sought to explore healthcare workers' perceived barriers and facilitators to engaging with expressive writing activities. We also sought to understand preferences with respect to writing activities, the mode of writing and the length

and duration of writing sessions. These questions are critically important for informing future feasibility studies and trials of expressive writing as an intervention to enhance the wellbeing of health and social care professionals, given the noted benefits of involving healthcare workers in the development of workplace wellbeing interventions [28]. Facilitators and barriers to engagement discussed by participants were predominantly aligned with the reflective motivation (46% of codes) and physical opportunity (31% of codes) sub-components of the COM-B model, with these two sub-components collectively accounting for over three quarters of all facilitators and barriers mentioned. With respect to reflective motivation, key subthemes highlighted that participants have a need to understand the benefits of expressive writing, that writing could be an effective debrief for healthcare workers and that intrinsic rewards may promote engagement. However, it was clearly expressed that healthcare workers face a range of structural workplace stressors which cannot be fully overcome via engagement with expressive writing alone. With respect to physical opportunity, key findings were that a conducive physical environment, with dedicated time to engage with expressive writing, are important considerations. Herein, we focus predominantly on the factors discussed by participants that are important considerations for the design of future evaluation studies.

**Writing format**

With respect to writing preferences, participants expressed a clear inclination for writing by hand rather than typing. This was an interesting finding, because given reports in the literature that there are no differences in outcomes for typing versus writing by hand (e.g., Layous, Katherine, & Lyubomirsky, 2013), there has been a recent trend in the expressive writing literature towards delivering expressive writing interventions online. Online delivery of expressive writing activities has several advantages for researchers and participants. This format makes it simple for researchers to ascertain adherence (e.g., that participants have completed the activities at the prescribed times) and avoids the need for transcription if analysis of the writing scripts is required. Some participants in the present study indicated that they always have an electronic device to hand, which they could use to write, whereas they don't always have a pen and paper. Given the stated preference for writing by hand, it would seem appropriate to allow participants the choice to write using writing by hand or typing (or a combination of both) in future evaluation studies of expressive writing in health and social care professionals.

With respect to the choice of writing activity, the three activities that were most strongly endorsed were three good things, written benefit finding and gratitude letter. These were the only activities to be endorsed by more than 50% of the participants. One of the clear barriers that emerged from the data was a lack of time to engage with the activities (see a more detailed discussion of this below). Perhaps a key reason that three good things was most strongly endorsed is that this was perceived as a relatively more straightforward and less time-consuming technique, aligned with a previous study investigating the feasibility of the three good things activity in healthcare workers [15]. Participants viewed both three good things and written benefit finding as useful techniques, because both techniques facilitate reframing – i.e., if an individual has had a stressful day at work, these techniques would enable them to refocus their mindset on positive outcomes from the day, or from a negative experience. Participants referred to written benefit finding as akin to a debrief, which they suggested would be beneficial during stressful periods at work or following exposure to an adverse event at work. Effective debriefs should serve a dual purpose of enhancing clinical care and supporting the psychological wellbeing of healthcare workers. However, it is known that their use is both inconsistent and infrequent [31]. Therefore, an easily self-administered writing technique may serve as a short-term, individualised form of debrief for processing challenging events that may help to alleviate work-related stress until a formal debrief with colleagues can be accessed. Indeed, single session debriefs following workplace traumas are thought to be ineffective [32], thus expressive writing could be used as an alternative, ongoing therapy, possibly as an adjunct to counselling or other psychological therapy. Participants frequently discussed the potential for expressive writing techniques to serve as a debrief, and the potential (workplace) wellbeing benefits as a reflective motivator to engaging with the techniques in this way. Similarly, participants outlined that using writing techniques such as written benefit finding as an alternative to talking about emotions would be a reflective motivator to engaging with these activities.

In terms of optimising psychological capability, participants commented that they were in favour of more structured techniques, with techniques such as three good things, written benefit finding and gratitude letter writing being clearer with respect to the requirements of the activities and how they should be structured. Techniques endorsed by fewer participants, such as writing about positive experiences or WED, provide less structured instructions and allow greater narrative freedom, although it was discussed that providing writing prompts or a template might make the activities easier. By contrast, participants also mentioned that writing instructions which are overly prescribed may interfere with spontaneous self-reflection, and some participants stated that they wouldn't want to be restricted to writing in a particular way or using a prescribed mode of writing. Participants pointed out that some flexibility is needed to allow people to engage with techniques that feel right for them, depending on individual differences and context. It was suggested that, especially in the early stages of adoption, people might find it difficult to engage with the activities, which highlights the importance of keeping the activities as simple as possible for participants to engage with, without making the instructions or format too complex. Many participants endorsed multiple activities, and some expressed a preference for having a variety of different writing techniques that they could use, in different contexts, rather than just being limited to using a single technique. An implication for future studies is that there could be some flexibility with the types of techniques that participants are permitted to use. This would perhaps make the experience less monotonous for participants and could promote reflective motivation. This is an important consideration for future trials, because a balance would need to be struck between giving participants flexibility to write in a way that suits them, but concurrently having a specified framework and instructions in place to promote scientific rigour.

With respect to writing duration and frequency, there was no clear consensus. Duration and frequency preferences tended to vary depending on the writing technique(s) that participants endorsed. Many participants who endorsed three good things suggested that they would engage with this activity more frequently, as they perceived that it would take less time and could be easily incorporated into the end of each working day or every evening before bed. Others suggested that they would prefer to engage more deeply but less frequently with expressive writing activities, spending more time exploring their emotions in written form, but that they would likely do this less often due to the time involved. Perhaps a key message here is that future evaluations need to provide some degree of flexibility with respect to the length and duration of expressive writing, to account for individual preferences. Alternatively, activities such as three good things, which could be engaged with at the end of each (working) day, may be a good candidate for future evaluation, on the basis of the preferences outlined by participants.

## Establishing a favourable environment for writing

Aligned with the physical opportunity sub-component of the COM-B model, participants expressed the importance of dedicating time and having a conducive environment to engage with expressive writing activities. It was unsurprising that participants frequently discussed a lack of time, both at work and outside of work, and difficulty establishing expressive writing into one's daily routine as key barriers to engagement. Therefore, participants in future evaluation studies would likely benefit from some support and guidance with forming a routine around expressive writing. Many participants highlighted that they would not have any time available in their working day to engage with expressive writing, and a substantial organisational culture shift within teams would likely be required to establish time within the working day to enable engagement at work. This would require support from managers, and potentially violates the ethos of many healthcare workers who may hold views that at work they should be prioritising the care of their patients over their own self-care. Participants suggested that support from their managers would be a key facilitator to engagement with writing were to take place in workplace settings. Therefore, when developing future intervention trials, it would likely be useful for researchers to seek the support and 'buy-in' from managers to promote the intervention. However, there is likely a balance to be struck here, because it is our view that if individuals perceive managers to be mandating engagement, this could adversely impact intervention efficacy. Unfortunately, no managers were included in our sample, so their perspectives on writing during working time or in workplace settings are hitherto unknown.

Relatedly, some participants stated that their home environment wouldn't be conducive, because they don't have a distraction-free location to write. Should future trials require participants to write at home, an important consideration would be to support participants to ensure they can secure a time and place to write where they would be free of distraction. As participants discussed that having a calm and private location to write would be important to them, it would perhaps be appropriate in future trials to discuss with participants where they intend to write, and to encourage participants to consider proactively any calm and private locations available to them where they could go to write routinely. This also aligns with the social opportunity sub-component of the COM-B model, as participants discussed the importance of having some quiet time and a space where they wouldn't be distracted by others to write. Involving friends and family in the activities could also be a way to promote engagement, as discussed by the participants.

### Reminders and prompts

With respect to psychological capability, participants reported that remembering to engage with expressive writing activities would be a potential barrier. Participants suggested that reminders would support regular engagement, but from an intervention implementation point of view, such reminders would ideally be led by the participants themselves [33]. Another possibility would be to support participants in a trial to embed expressive writing activities into their daily routine. Participants highlighted the importance of making expressive writing habitual to support regular engagement. A further reason that participants favoured brief techniques such as three good things, was that it was perceived that this technique could be completed more quickly or more easily embedded into an individual's routine enabling the act of writing to become habitual. Forming a habit is known to be a key facilitator of behaviour change, and can bridge the gap between intending to engage with the behaviour and performing the behaviour [34]. Thus, strategies to promote habit formation should be considered as part of future expressive writing trials. Participants also felt that the more familiar they become with the techniques, the easier they would find them, which fits with the notion that familiarity and forming a routine around writing activities is key to engagement.

### Perceived benefits

Perhaps unsurprisingly, a clear reflective motivator was perceiving the benefits of engaging with the activities. Participants stated that if they were not enjoying the process of expressive writing or had a general dislike for writing, they would not be willing to engage. Participants also stated that they could potentially find some of the techniques discomforting or stressful to engage with, and that there was a chance that having to regularly engage with expressive writing could feel like a chore. These are important considerations for future expressive writing trials. It is possible that some of the reflective motivation facilitators highlighted by participants could overcome some of these barriers, such as knowledge of evidence supporting benefits, monitoring benefits, and rewards. For example, if participants perceive a benefit from engaging, then perhaps some of these barriers would be overcome naturally. Participants also discussed that being explicitly informed about the potential benefits of expressive writing would motivate them to engage with the activities. This is an interesting perspective, given that in research studies investigating the efficacy of expressive writing, the potential benefits of the techniques are often not disclosed to participants to minimise demand characteristics [21]. Although disclosing this information to participants could inflate any effects of expressive writing [21,35], this will act as a reflective motivator to engagement and maximise intervention success. A further interesting idea for consideration in future trials which participants floated is to incorporate self-report mood measures to allow participants to explicitly see an improvement in their mood over time. Furthermore, participants endorsed the notion of including tangible rewards, such as shopping vouchers, to incentivise engagement with the activities. Implementing such incentives would need to be considered cautiously, because it is well known that implementation of such extrinsic motivators can lead to participants disengaging from interventions in the longer term, when the extrinsic motivator is removed [36]. Finally, participants stated that some occupational stressors are so severe that it is unlikely expressive writing could overcome the adverse effects of these. For this

reason, we suggest that expressive writing should not be advocated as the sole approach to addressing psychological wellbeing among health and social care workers. Instead, it should be viewed as one of a suite of possible approaches that could be used.

## Addressing the barriers

Barriers relating to psychological capability, automatic and reflective motivation, and social and physical opportunity could be mapped onto the following intervention functions within the Behaviour Change Wheel [27]: environmental restructuring, education, and enablement. It is proposed that these functions should be targeted using specific behaviour change techniques [BCTs; 37] to facilitate engagement in expressive writing among healthcare workers. For example, environmental restructuring could include adding objects to the environment (12.5) or prompts/cues (7.1) such as digital reminders for staff to complete the writing. Indeed, prompts were identified by participants as a key facilitator for automatic motivation. For education, staff could be provided information about health consequences (5.1) from a credible source (9.1). This could enhance beliefs regarding positive outcomes from engaging in expressive writing. To increase enablement, staff could be provided with materials to allow self-monitoring of outcome(s) of behaviour (2.4), which was also suggested as a facilitator by participants. These are just a few BCTs which could be targeted to facilitate engagement in expressive writing among healthcare workers. However, it would be useful to conduct further research exploring barriers and facilitators to intervention implementation among staff in supervisory, management and leadership roles. This would allow a more comprehensive application of the Behaviour Change Wheel to implement expressive writing within healthcare settings.

## Limitations

While this study has enabled us to develop a comprehensive understanding of the preferences, facilitators and barriers of healthcare workers to engaging with expressive writing activities, there are some key limitations. The oldest participant was 44, with the other ten participants aged in their 20s and 30s. Thus, this is a relatively young sample at the early stages of their career. Not only does this mean that we have failed to capture the perspectives of relatively older participants, but we may also have failed to recruit participants who have been exposed to work-related stress over a longer period and thus may be at a greater risk of severe burnout. Further, all participants work in the NHS. Although the NHS is the largest employer in the UK [38], we haven't obtained the perspective of healthcare professionals working in the private sector, nor other international health services. All participants were from London or North East England, and the majority of participants were drawn from the psychological workforce. This is potentially problematic because participants working in psychological health likely have a unique perspective on expressive writing interventions, and some commented that they use activities such as these as part of their routine clinical practice. The recruitment strategy employed meant that we were unable to recruit a stratified sample on the basis of cultural background, so we are not able to draw any conclusions with respect to whether the views expressed transcend the diverse cultural groups from which healthcare workers may be drawn. Further, the study advert may have appealed more to individuals who are inclined to be more naturally reflective or have experience with expressive writing, thus, the sample may be biased towards individuals who are most likely to endorse the techniques. The sample was missing some key frontline healthcare roles, including nursing and frontline emergency medicine, which are known to be particularly stressful occupations [39]. Ideally, the sample would have included some participants in managerial roles. It would have been useful to capture the unique perspective of individuals working in such roles on the basis that i) the demands of such roles are intrinsically stressful [40], and ii) managers would provide useful insight on such factors as whether it would be logistically possible to enable workers within the service they manage to engage with expressive writing activities during working hours. The lack of diversity in the sample potentially minimises the generalisability of the findings. While the sample size was adequate, the sample comprised only 11 participants, and recruiting a larger sample may have facilitated the inclusion of a more diverse range of individuals and

captured a more diverse range of views. Finally, it is acknowledged that the present study relates to participants' perceptions of expressive writing, rather than actual experience. This study is viewed as an initial and crucial step towards developing expressive writing interventions for healthcare workers, in that it is important to understand potential barriers and facilitators to inform intervention development. However, barriers and facilitators reported by participants with experience of engaging with these activities may differ from some of those reported here. Future qualitative studies should determine the barriers and facilitators of participants following a period of engagement with the activities.

## Conclusions

In conclusion, we sought here to better understand the perceived preferences, facilitators and barriers of health and social care workers to engaging in expressive writing activities. This is important work to inform future feasibility and definitive trials of expressive writing in this group, because the design of future trials must incorporate the views of participants to ensure the maximum likelihood of engaging with and the acceptability of the intervention. A key learning was that participants have a clear preference for three good things, written benefit finding and gratitude letter techniques. The former was preferred, as participants perceived that this is a relatively brief technique that could be most easily incorporated into a daily routine, allowing dedicated time for writing in alignment with the physical opportunity sub-component of the COM-B model. Participants also felt that these three techniques would be most beneficial for improving their wellbeing. Participants expressed a clear preference for writing by hand over typing, and there was no clear consensus with respect to the frequency with which participants would want to engage, which was to some extent dependent on the specific techniques they preferred to use. There were important findings with respect to facilitators and barriers to engagement that are key considerations with respect to the success of future trials. Participants highlighted that techniques should be simple, and that support would be required, from either managers, researchers or friends and family, to allow them to successfully embed the techniques into their daily routine. Generally, participants expressed the view that they would need some flexibility to enable them to engage with the activities, so a consideration is that future evaluation studies should try to avoid being too prescriptive in terms of precise writing instructions, or the time and place that writing should occur. By embedding the perceived facilitators and attempting to remove some of the barriers highlighted by participants, the success of future expressive writing evaluation studies with healthcare workers is likely to be optimised. Expressive writing has potential for supporting the wellbeing on healthcare workers, potentially as one component of a whole system approach with initiatives at sector, organisational, team and individual levels.

## Author contributions

**Conceptualization:** Michael Smith, Nicola O'Brien, Mark A. Wetherell.

**Formal analysis:** Michael Smith, Lauren M Hoult.

**Investigation:** Lauren M Hoult.

**Methodology:** Michael Smith, Daniel Rippon.

**Project administration:** Michael Smith, Lauren M Hoult.

**Resources:** Michael Smith.

**Supervision:** Michael Smith.

**Writing – original draft:** Michael Smith, Lauren M Hoult.

**Writing – review & editing:** Lauren M Hoult, Daniel Rippon, Nicola O'Brien, Dawn Branley-Bell, Lucie Byrne-Davis, Caitriona Collins, Stephen Gallagher, Gail Kinman, Arron Mark, Daryl B. O'Connor, Kavita Vedhara, Glenn P. Williams, Mark A. Wetherell.

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
