## [Decision Letter · Decision Letter 0]

15 Apr 2025

Dear Dr. Smith,

Thank you for submitting your manuscript to PLOS ONE. After careful consideration, we feel that it has merit but does not fully meet PLOS ONE’s publication criteria as it currently stands. Therefore, we invite you to submit a revised version of the manuscript that addresses the points raised during the review process.

We look forward to receiving your revised manuscript.

Kind regards,

Fatma Refaat Ahmed, Ph.D.

Academic Editor

PLOS ONE

Journal Requirements:

“I have read the journal's policy and the authors of this manuscript have the following competing interests: Michael A. Smith and Mark A.Wetherell have run expressing writing workshops which have generated income to Northumbria University.”

Reviewers' comments:

Reviewer's Responses to Questions

**Comments to the Author**

1. Is the manuscript technically sound, and do the data support the conclusions?

Reviewer #1: Yes

Reviewer #2: Yes

2. Has the statistical analysis been performed appropriately and rigorously?

Reviewer #1: N/A

Reviewer #2: Yes

3. Have the authors made all data underlying the findings in their manuscript fully available?

Reviewer #1: Yes

Reviewer #2: Yes

4. Is the manuscript presented in an intelligible fashion and written in standard English?

Reviewer #1: Yes

Reviewer #2: Yes

Reviewer #1: I thank the authors for the opportunity to review this work. I enjoyed reading the manuscript and commend the considerable effort they have put into it. Overall, the paper is well-written and has the potential to make a meaningful contribution to understanding healthcare and social care professionals' perspectives on expressive writing, spanning Pennebaker's Writing to Expressive Disclosure (WED) paradigm and activities rooted in positive psychology. However, I have some reservations regarding the methods and the generalizability of the findings. I believe the authors could strengthen their contributions by addressing the following questions and suggestions.

MAJOR COMMENTS

- Motivation. Authors point out that there has been little research investigating expressive writing for healthcare staff, but Sexton and Adair's work is a promising example for introducing 3GT into the domain. I think the authors can argue a better case as to why they need to introduce a range of expressive writing activities to these busy healthcare professionals. I see 6 in total from the slide deck, but 5 seem to be from Positive Psychology and the other Pennebaker's. Authors briefly discuss the focus on negative vs positive emotions from the two broad categories, but I think the paper can benefit from a more deeper discussion into how the different approaches align with the unique roles and responsibilities of healthcare professionals face in their workplaces. Additionally, providing more context about the nature of the target population's work would help unfamiliar readers understand how expressive writing could realistically fit into their routines.

- Methods. I have several questions about the overall study design and execution. (1) Was there a specific reason for opting for a small qualitative study? While interviews can offer rich insights, a survey might have allowed for broader participation and yielded complementary findings. The authors should clarify why a qualitative approach was necessary. (2) The study appears to rely on participants' hypothetical perceptions of the expressive writing activities rather than their actual experiences. Why were participants not asked to engage in these activities themselves? The absence of direct experience limits the findings to anticipated concerns and benefits, rather than practical strategies or challenges. Perhaps introducing a hypothetical example, such as asking participants to imagine completing a written benefit-finding exercise during a typical workday, could have been an option to discuss "how often and how long" questions authors ask in the beginning. I'd appreciate if the authors could give more justifications on how such hypothetical perceptions can reliably inform barriers and facilitators for integrating these activities into the workplace. (3) Table 1 should include information about participants' prior familiarity with the expressive writing activities described in the slide deck. Since the deck consisted of 10 slides with brief descriptions of each technique, any prior experience with these activities should be noted to mitigate potential bias. (4) As noted in the Limitations, the participant group may not represent broader cultural or professional diversity, which may introduce bias. The authors should explain why a more diverse or stratified sample was not feasible and discuss whether their findings can be generalized despite this limitation. (5) While the thematic analysis seems broadly aligned with the COM-B model's subconstructs, the final themes should be presented in the manuscript, either in text or as a table/figure.

- Discussion. This section could be strengthened by reducing reiteration of findings and delving deeper into their implications. For example, could habitual engagement with expressive writing activities feasibly benefit healthcare professionals? Given the structural and administrative constraints identified as barriers, insights from participant interviews could help readers understand how these activities might be sustained or redesigned to meet the unique needs of healthcare professionals.

MINOR COMMENTS

- A description or table summarizing the expressive writing activities used in the study would aid readers, even if available via the OSF link.

- A thorough proofreading of the manuscript would address the minor typos or misspellings noted.

Reviewer #2: Impression: This qualitative (psychology) study assumed that Expressive writing might be an effective intervention for reducing stress and enhancing wellbeing in the healthcare workers. The study design, and thematic analysis are excellent. There are a few points for value addition that need correction or clarification.

The sample size was relatively very small (11), although researchers tried to convince (in methods) that even 10 is enough for a qualitative study design. This issue should be there in limitations towards the end of the discussion.

It was not clear if these health care workers were already having stress or wellbeing issues. This would be a definite confounder if this factor was not addressed.

There are too many mentions of LH and MS in the methods. The initials can be easily avoided by rewriting the manuscript in routine passive voice. That would avoid confusion about methods.

Statement “Length of service ranged between two months and 8 years” table-1: One wonders if it was a uniform cohort. In addition please note that table-1 reflects participant:3 with experience of just 2 weeks (with 3 years PWP) while there is no participant with 8 years experience.

Participants mentioned in methods are 11 (line 5) as well as line 5 under heading participants in methods. Please recheck if 10 is written correctly at a place in methods

Under the heading Reflexive statements: The first paragraph suits better for limitations (at the end of discussion). In addition, after a remark about collaboration between participants and authors, what is the advantage of giving such a long introduction of two of the researchers (MS and LH)?

In the sentence “The majority of participants stated that their preferred mode of writing would be handwriting”: Consider replacing “handwriting” with “writing by (or with) hand” (optional). Please note that it is at least at 12 more places (like in Discussion, under writing format)

References: Some of the references look incomplete (examples: 6, 7, 12) Please recheck

**Do you want your identity to be public for this peer review?** For information about this choice, including consent withdrawal, please see our Privacy Policy

Reviewer #1: No

Reviewer #2: No

---

## [Author Response · Author response to Decision Letter 1]

16 May 2025

We thank the reviewers for the very insightful feedback received. It is our opinion that incorporating the suggested amendments into the manuscript has much improved the paper.

Please note that amendments to the text are marked-up using tracked changes, as requested.

Reviewer 1

Authors point out that there has been little research investigating expressive writing for healthcare staff, but Sexton and Adair's work is a promising example for introducing 3GT into the domain. I think the authors can argue a better case as to why they need to introduce a range of expressive writing activities to these busy healthcare professionals. I see 6 in total from the slide deck, but 5 seem to be from Positive Psychology and the other Pennebaker's. Authors briefly discuss the focus on negative vs positive emotions from the two broad categories, but I think the paper can benefit from a more deeper discussion into how the different approaches align with the unique roles and responsibilities of healthcare professionals face in their workplaces. Additionally, providing more context about the nature of the target population's work would help unfamiliar readers understand how expressive writing could realistically fit into their routines.

We thank the reviewer for these suggestions and the opportunity to strengthen the rationale. The rationale for the inclusion of these six activities in the slide deck has now been added to the Procedure section. We considered it important to include Pennebaker’s ‘flagship’ WED activity, and identified a further five positive expressive writing activities from a recent systematic review. Additionally, we have added a new sentence to the introduction to extend the rationale with respect to why we think these techniques may be beneficial for healthcare workers: “WED may offer healthcare workers the opportunity to process stressful or difficult events which they experience during their working lives, while positive expressive writing activities may provide an opportunity for reframing and focusing on positive aspects of their roles.”. With respect to how the activities could fit into the routines of healthcare workers, this was a concern we had when embarking on this work and was a key motivation for running the study. We are very keen to ensure that future trials don’t place an unrealistic burden on these busy professionals. We mention this in the Intro: “Feasibility studies have shown that there are issues with written benefit finding for informal carers (18), and positive writing for professional carers (26) in terms of participants reporting having a lack of time to engage with expressive writing. However, qualitative comments reported by healthcare workers in Sexton and Adair’s (15) study convey an overwhelming enthusiasm with the three good things format, suggesting that this briefer exercise may be preferred by participants as it is perceived as a more feasible technique to engage with.”

Was there a specific reason for opting for a small qualitative study? While interviews can offer rich insights, a survey might have allowed for broader participation and yielded complementary findings. The authors should clarify why a qualitative approach was necessary.

We thank the reviewer for highlighting that our justification for a qualitative approach was a little opaque. We anticipated that the barriers and facilitators which participants would identify would be diverse and complex (and our data demonstrate that this assumption was correct!). It is our view that we wouldn’t have been able to gain such a deep understanding of these complex issues from a quantitative approach, and it was important to gain a deep understanding in order to inform future intervention studies. For this reason, a qualitative approach was most appropriate. We have now added a justification for the qualitative approach in the Design section: “This method was chosen to enable an in-depth exploration of participants’ barriers and facilitators to engaging with expressive writing, and to gain an understanding of individuals’ subjective experiences to ascertain how best to deliver an expressive writing intervention to healthcare workers”.

The study appears to rely on participants' hypothetical perceptions of the expressive writing activities rather than their actual experiences. Why were participants not asked to engage in these activities themselves? The absence of direct experience limits the findings to anticipated concerns and benefits, rather than practical strategies or challenges. Perhaps introducing a hypothetical example, such as asking participants to imagine completing a written benefit-finding exercise during a typical workday, could have been an option to discuss "how often and how long" questions authors ask in the beginning. I'd appreciate if the authors could give more justifications on how such hypothetical perceptions can reliably inform barriers and facilitators for integrating these activities into the workplace.

We agree with the reviewer that this is a limitation of the work, and have added the following to the Discussion to acknowledge this: “Finally, it is acknowledged that the present study relates to participants’ perceptions of expressive writing, rather than actual experience. This study is viewed as an initial and crucial step towards developing expressive writing interventions for healthcare workers, in that it is important to understand potential barriers and facilitators to inform intervention development. However, barriers and facilitators reported by participants with experience of engaging with these activities may differ from some of those reported here. Future qualitative studies should determine the barriers and facilitators of participants following a period of engagement with the activities.”

Table 1 should include information about participants' prior familiarity with the expressive writing activities described in the slide deck. Since the deck consisted of 10 slides with brief descriptions of each technique, any prior experience with these activities should be noted to mitigate potential bias.

We agree that this is useful information to include. We have updated the participants section of the manuscript as well as Table 1 to detail participants’ level of familiarity with expressive writing techniques.

As noted in the Limitations, the participant group may not represent broader cultural or professional diversity, which may introduce bias. The authors should explain why a more diverse or stratified sample was not feasible and discuss whether their findings can be generalized despite this limitation.

We have added a sentence to the Discussion to highlight that the lack of diversity in the sample potentially minimises the generalisability of the findings. We have detailed in the manuscript that participants were recruited via social media adverts and the researchers’ networks. We explain that this sampling strategy meant that it wasn’t possible to obtain a stratified sample.

While the thematic analysis seems broadly aligned with the COM-B model's subconstructs, the final themes should be presented in the manuscript, either in text or as a table/figure.

The themes are presented in Table 3 and Figure 5.

The Discussion could be strengthened by reducing reiteration of findings and delving deeper into their implications. For example, could habitual engagement with expressive writing activities feasibly benefit healthcare professionals? Given the structural and administrative constraints identified as barriers, insights from participant interviews could help readers understand how these activities might be sustained or redesigned to meet the unique needs of healthcare professionals.

We make several suggestions throughout the Discussion relating to the implications of the findings for the design of future intervention trials. These include: giving participants choice and flexibility with respect to the mode of writing, writing activity and writing frequency; that brief techniques, particularly those which involve reframing, may be most effective; that expressive writing could be used as an adjunct debrief; supporting participants with establishing a routine to implement habitual expressive writing; and providing prompts. We also highlight the potential for expressive writing for supporting the wellbeing on healthcare workers (see e.g. final sentence of Conclusions).

A description or table summarizing the expressive writing activities used in the study would aid readers, even if available via the OSF link.

We thank the reviewer for this suggestion. We have added a new table (Table 2) to summarise the expressive writing activities presented to the participants.

A thorough proofreading of the manuscript would address the minor typos or misspellings noted.

We have undertaken a thorough proofread of the manuscript and amended to improve clarity, grammar, spelling errors and typos.

Reviewer 2

The sample size was relatively very small (11), although researchers tried to convince (in methods) that even 10 is enough for a qualitative study design. This issue should be there in limitations towards the end of the discussion.

We have added a sentence to the Discussion to highlight this as a limitation: “While the sample size was adequate, the sample comprised only 11 participants, and recruiting a larger sample may have facilitated the inclusion of a more diverse range of individuals and captured a more diverse range of views.”

It was not clear if these health care workers were already having stress or wellbeing issues. This would be a definite confounder if this factor was not addressed.

We asked participants to discuss whether they were currently experiencing any work or non-work stressors. One participant was on leave from work due to stress. Additionally, seven participants reported current stress due to overwork (admin, high turnover of patients, lack of time for breaks), seven participants reported stress due to role responsibilities (identifying and managing risk, managing expectations, decision making, dealing with patient complexities), one participant reported stress due to low pay and one participant reported stress due to there being no mechanisms for debrief or reflection following incidents at work. We have added these details to the Results section and also include the activities participants’ report using to cope with stress.

There are too many mentions of LH and MS in the methods. The initials can be easily avoided by rewriting the manuscript in routine passive voice. That would avoid confusion about methods.

We have rewritten parts of the method section in the passive voice to reduce the number of instances that LH and MS are mentioned.

Statement “Length of service ranged between two months and 8 years” table-1: One wonders if it was a uniform cohort. In addition please note that table-1 reflects participant:3 with experience of just 2 weeks (with 3 years PWP) while there is no participant with 8 years experience.

Thank you for highlighting the lack of clarity with the way we had reported lack of service data. This was complicated by the fact that some participants had changed job roles within the health service, and the way we reported this was admittedly unclear, as the reviewer suggests. For clarity, we have now amended Table 1 so that only participants’ total length of service is displayed. We acknowledge in the Discussion that this is not a homogenous group.

Participants mentioned in methods are 11 (line 5) as well as line 5 under heading participants in methods. Please recheck if 10 is written correctly at a place in methods

Our initial aim was to recruit 10 participants. We have clarified this at the relevant point of the Methods section.

Under the heading Reflexive statements: The first paragraph suits better for limitations (at the end of discussion). In addition, after a remark about collaboration between participants and authors, what is the advantage of giving such a long introduction of two of the researchers (MS and LH)?

The reflexive statements we provide are aligned with Consolidated Criteria for Reporting Qualitative Studies (COREQ) guidelines. We provide statements only for the two researchers who were primarily responsible for undertaking the analyses, as it would be excessive to provide this for all authors. Additionally, the biases of the two authors who took primary responsibility for the analyses are most likely to influence the interpretation. We have added a sentence to clarify this.

In the sentence “The majority of participants stated that their preferred mode of writing would be handwriting”: Consider replacing “handwriting” with “writing by (or with) hand” (optional). Please note that it is at least at 12 more places (like in Discussion, under writing format)

We thank the reviewer for this suggestion. We have replaced the term ‘handwriting’ with ‘writing by hand’ throughout the manuscript, and in Figure 3.

References: Some of the references look incomplete (examples: 6, 7, 12) Please recheck

These references have been updated.

---

## [Decision Letter · Decision Letter 1]

8 Jul 2025

Facilitators and barriers to engaging in expressive writing among health and social care professionals

PONE-D-24-49111R1

Dear Dr. Smith,

We’re pleased to inform you that your manuscript has been judged scientifically suitable for publication and will be formally accepted for publication once it meets all outstanding technical requirements.

Kind regards,

Fatma Refaat Ahmed, Ph.D.

Academic Editor

PLOS ONE

Additional Editor Comments (optional):

Reviewers' comments:

Reviewer's Responses to Questions

**Comments to the Author**

Reviewer #2: All comments have been addressed

Reviewer #3: All comments have been addressed

2. Is the manuscript technically sound, and do the data support the conclusions?

Reviewer #2: Yes

Reviewer #3: Yes

3. Has the statistical analysis been performed appropriately and rigorously?

Reviewer #2: N/A

Reviewer #3: Yes

4. Have the authors made all data underlying the findings in their manuscript fully available?

Reviewer #2: Yes

Reviewer #3: Yes

5. Is the manuscript presented in an intelligible fashion and written in standard English?

Reviewer #2: Yes

Reviewer #3: Yes

Reviewer #2: Thank you for addressing all my suggestions. I think some of the figures are duplication of the results and may not be needed.

Reviewer #3: I appreciate the opportunity to review your revised manuscript. The comments and recommendations by the reviewers have been successfully addressed. Exploring the facilitators and barriers to engaging in expressive writing by healthcare and social service providers adds to the science of self care among this population.

**Do you want your identity to be public for this peer review?** For information about this choice, including consent withdrawal, please see our Privacy Policy

Reviewer #2: No

Reviewer #3: No

---

## [Editor Report · Acceptance letter]

PONE-D-24-49111R1

PLOS ONE

Dear Dr. Smith,

I'm pleased to inform you that your manuscript has been deemed suitable for publication in PLOS ONE. Congratulations! Your manuscript is now being handed over to our production team.

Kind regards,

on behalf of

Dr. Fatma Refaat Ahmed

Academic Editor

PLOS ONE